# Ten-year in-hospital mortality trends among Japanese injured patients by age, injury severity, injury mechanism, and injury region: A nationwide observational study

Chiaki Toida[1,2]*, Takashi Muguruma[2], Masayasu Gakumazawa[2], Mafumi Shinohara[2], Takeru Abe[2], Ichiro Takeuchi[2]

1 Department of Disaster Medical Management, The University of Tokyo, Tokyo, Japan, 2 Department of Emergency Medicine, Yokohama City University Graduate School of Medicine, Yokohama, Japan

* toida-ygc@umin.ac.jp

## Abstract

The Injury Severity Score (ISS) is widely used in trauma research worldwide. An ISS cutoff value of ≥16 is frequently used as the definition of severe injury in Japan. The mortality of patients with ISS ≥16 has decreased in recent years, owing to the developing the trauma care system. This study aimed to analyze the prevalence, in-hospital mortality, and odds ratio (OR) for mortality in Japanese injured patients by age, injury mechanism, injury region, and injury severity over 10 years. This study used the Japan Trauma Data Bank (JTDB) dataset, which included 315,614 patients registered between 2009 and 2018. 209,290 injured patients were utilized. This study evaluated 10-year trends of the prevalence and in-hospital mortality and risk factors associated with in-hospital mortality. The overall in-hospital mortality was 10.5%. During the 10-year study period in Japan, the mortality trend among all injured patient groups with ISS 0–15, 16–25, and ≥26 showed significant decreases (p <0.001). Moreover, the mortality risk of patients with ISS ≥26 was significantly higher than that of patients with ISS 0–15 and 16–25 (p <0.001, OR = 0.05 and p<0.001, OR = 0.22). If we define injured patients who are expected to have a mortality rate of 20% or more as severely injured, it may be necessary to change the injury severity definition according to reduction of trauma mortality as ISS cutoff values to ≥26 instead of ≥16. From 2009 to 2018, the in-hospital mortality trend among all injured patient groups with ISS 0–15, 16–25, and ≥26 showed significant decreases in Japan. Differences were noted in mortality trends and risks according to anatomical injury severity.

## Introduction

Injury has been a major cause of death in Japan over the past few decades [1]. The Japan Trauma Data Bank (JTDB) was established to improve the quality of trauma care by collecting and analyzing data from injured patients [2, 3]. As direct comparison of the outcome of

**Data Availability Statement:** All relevant data are within the paper and its Supporting Information files.

**Funding:** C.T. received a grant from the General Insurance Association of Japan [Grant No.21-08]. The funders had no role in this study design, data collection and analysis, decision to publish, or preparation of the manuscript.

**Competing interests:** The authors have declared that no competing interests exist.

injured patients with different severities of injury is often noninformative and misleading, injury severity is an important factor to be considered when analyzing mortality and morbidity based on a nationwide trauma registry [4]. Therefore, it is essential to consider risk-adjusted outcome measurements such as trauma scores while using a nationwide database, which includes patients with various severities of injury [4, 5].

Trauma scores were developed to describe the injury severity or prognosis of injured patients with a single numerical value. Injury Severity Score (ISS) is the most widely used trauma score in trauma research worldwide [4–6]. As the ISS score is calculated based on an anatomical injury severity and correlates well with the mortality of injured patients, an ISS cut-off value of ≥16 was chosen as the definition of severe injury with high mortality risk based on the Major Trauma Outcome Study (MTOS) from 1982 to 1987, because patients with ISS ≥16 had an expected mortality rate of more than 20% [6–9]. Epidemiological trauma outcome research based on the JTDB data also frequently used an ISS cutoff value of ≥16 as the definition of severe injury [10–12].

The mortality of patients with trauma has decreased in recent years due to the development of the trauma care system [9–11]. Moreover, the mortality trend and risk of severely injured patients vary widely according to age, injury mechanism, injury region, and/or injury severity [3, 10–13]. However, to the best of our knowledge, no long-term study has evaluated the mortality trends and risks of injured patients in a Japanese cohort using detailed classification of age and/or injury severity. Therefore, this 10-year nationwide study aimed to analyze the prevalence, in-hospital mortality, and odds ratio (OR) for mortality in Japanese injured patients by age and injury severity including injury mechanism and injury region.

## Materials and methods

### Study setting and patient population

This retrospective observational nationwide study used data from the JTDB, which included data on demographic characteristics, comorbidities, means of transportation, injury mechanism, injury region indicated by the abbreviated injury scale (AIS) score, vital signs, prehospital/in-hospital procedures, and clinical outcomes. In 2018, there were 280 participating hospitals in all 47 prefectures in Japan [2]. The Japan Association for the Surgery of Trauma permitted open access and updating of existing medical information, and the Japan Association for Acute Medicine evaluated the submitted data [2].

This study used the JTDB dataset, which included 315,614 patients registered between January 1, 2009, and December 31, 2018. The inclusion criteria for this study were injured patients who were transferred from the scene of injury by an ambulance car and/or helicopter. Patients with burns and missing data on age, gender, injury mechanism, ISS, and/or survival outcome were excluded from this study. Fig 1 presents a flow diagram of the patient selection in this study and S1 Table presents number of patients with missing data by study year.

### Ethics statement

The study was conducted in accordance with the guidelines of the Declaration of Helsinki and approved by the Institutional Ethics Committee of Yokohama City University Medical Centre (approval no. B170900003). The approval for data access was provided by the Japanese Association for the Surgery of Trauma (Trauma Registry Committee). The requirement for informed consent from the patients was waived due to the observational nature of the study design.

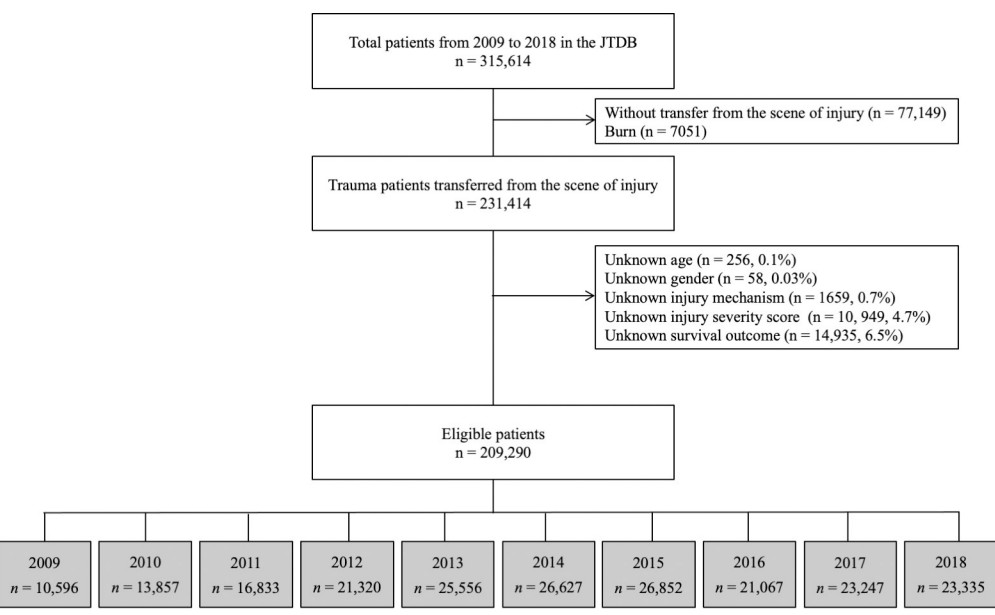

**Fig 1. Flow diagram of the study patient selection for this study.**

## Data collection and outcome measurements

We collected information on the following variables from the JTDB: demographic characteristics (age in years, gender, year of hospital admission); clinical parameters (injury mechanism, AIS of the injured region, ISS, Glasgow Coma Scale (GCS) score, systolic blood pressure (sBP), and heart rate (HR) at hospital admission); and outcomes (in-hospital mortality). The outcome measurements included the proportion of patients and in-hospital mortality according to the 10 groups categorized based on the hospital admission year from 2009–2018 split into nine age groups (0–4, 5–14, 15–24, 25–34, 35–44, 45–54, 55–64, 65–74, and ≥75 years old), two groups based on injury mechanism (blunt, penetrating injury), nine groups based on two or more injury regions with AIS ≥3 (polytrauma, head injury, facial injury, neck injury, chest injury, abdominal and pelvic injury, spinal injury, upper extremity injury, and lower extremity injury), and three groups based on injury severity (ISS 0–15, 16–25, and ≥26), which were classified according to previous studies [3, 6, 10–13].

## Statistical analysis

This study evaluated the following: (1) 10-year trends in prevalence and in-hospital mortality by age, injury mechanism, and injury region with two or more injuries with AIS ≥3; and (2) risk factors associated with in-hospital mortality over 10 years. In the primary analysis, the Jonckheere-Terpstra trend test was used to test for trends in continuous variables, and the Cochran-Armitage trend test was used to test for trends in categorical variables by hospital admission year. In the secondary analysis, the OR (95% confidence interval [CI]) for in-hospital mortality was calculated using a logistic regression model. The following variables were included in the multivariable logistic regression analyses: admission year, age, injury mechanism, two or more injury regions with AIS ≥3, and ISS. The dependent variable in the multivariable logistic regression analysis was in-hospital mortality. The results were expressed as medians and interquartile ranges (IQR, Q1–Q3) for continuous variables, and patient numbers and percentages for categorical variables. All statistical analyses were performed using the

STATA/SE software (version 17.0; StataCorp; College Station, TX, USA). Statistical significance was defined as a two-tailed P-value <0.05.

## Results

During the 10-year study period, we utilized data from a total of 209,290 injured patients (Fig 1). These patients were categorized into the following age groups: 0–4 years ($n$ = 2021, 1%); 5–14 years ($n$ = 7519, 4%); 15–24 years ($n$ = 20,909, 10%); 25–34 years ($n$ = 16,017, 8%); 35–44 years ($n$ = 19,575, 9%); 45–54 years ($n$ = 21,674, 10%); 55–64 years ($n$ = 26,866, 13%); 65–74 years ($n$ = 33,898, 16%); and ≥75 years ($n$ = 60,811, 29%). The number of patients with ISS 0–15, 16–25, and ≥26 was 1,18,547 (57%), 57,745 (28%), and 32,998 (16%), respectively. The median age and ISS score were 61 years (IQR, 38–77) and 13 (IQR, 9–21), respectively. The median GCS score, sBP, and HR at hospital admission was 15 (IQR, 13–15), 135 (IQR, 114–157), and 20 (IQR, 16–24), respectively. The overall in-hospital mortality rate was 10.5%.

The 10-year in-hospital mortality trends of all injured patients are shown in Table 1. In the Cochran-Armitage trend test, the in-hospital mortality of all injured patients significantly decreased from 13.7% in 2009 to 9.1% in 2018 (p <0.001). Similarly, the in-hospital mortality rates in patients with ISS 0–15, 16–25, and ≥26 showed significant decrease (from 2.5% to 1.8%, p = 0.000; from 15.6% to 10.4%, p <0.001; from 43.1% to 35.0%, p <0.001, respectively; Fig 2).

The in-hospital mortality trends among injured patients by age group, injury mechanism, and injury region according to three groups based on injury severity (ISS 0–15, 16–25, and ≥26) are shown in S2–S4 Tables. Among injured patients with age >15 years, in-hospital mortality of patients with ISS 16–25 and ≥26 significantly decreased over the 10-year study period (S2 Table). The in-hospital mortality among patients with blunt injury showed a significant decrease in all ISS groups (S3 Table). Moreover, in patients with ISS from 16–25, the in-hospital mortality of patients with polytrauma, head, chest, abdominal and pelvic, spinal, upper extremity, and lower extremity injuries with AIS ≥3 showed a significant decrease (S4 Table).

Table 2 shows the results of multivariable logistic analyses. The mortality risk of patients admitted in 2018 (comparative controls) was significantly lower than that of patients admitted before 2016. The mortality risk of patients aged ≥76 years was significantly higher than that of patients in other age categories. Patients with polytrauma, head, neck, chest, abdominal and pelvic, and lower extremity injury with AIS ≥3 were significantly associated with higher OR of in-hospital mortality. Moreover, the mortality risk of patients with ISS ≥26 was significantly higher than that of patients with ISS 0–15 and 16–25 (p <0.001, OR = 0.05, 95%CI = 0.045–0.051 and p <0.001, OR = 0.22, 95%CI = 0.206–0.224, respectively).

## Discussion

This 10-year nationwide study in Japan showed that the in-hospital mortality trend significantly decreased in all injured patient groups with ISS 0–15, 16–25, and ≥26. However, there were differences in the mortality trends and risk according to the age, injury mechanism, injury region, and anatomical severity. Moreover, the in-hospital mortality and OR for mortality in the patient group with ISS ≥26 were higher than those in two patient groups with ISS 0–15 and 16–25.

There were significant differences in the mortality risk of injured patients according to the severity of anatomical injury. Previous studies have shown that the mortality of injured patients with ISS ≥16 has been decreasing to below 20% [9–11, 13]. This study also showed that the mortality rate of injured patients with ISS 0–15 and 16–25 in 2018 was 20% or less, regardless of age, injury mechanism, and injury region. Moreover, the OR for mortality of

**Table 1. Demographics and characteristics of injured patients by year groups.**

| Variables | 2009 | 2010 | 2011 | 2012 | 2013 | 2014 | 2015 | 2016 | 2017 | 2018 | p-value |
|---|---|---|---|---|---|---|---|---|---|---|---|
| | n = 10,596 | n = 13,857 | n = 16,833 | n = 21,320 | n = 25,556 | n = 26,627 | n = 26,852 | n = 21,067 | n = 23,247 | n = 23,335 | |
| Male, n (%) | 6869 (65) | 9046 (65) | 10,894 (64) | 13,595 (64) | 16,067 (63) | 16,833 (63) | 16,982 (63) | 13,313 (63) | 14,621 (63) | 14,520 (62) | <0.001 |
| Age in year, (median, IQR) | 56 (32–73) | 56 (32–73) | 58 (35–74) | 60 (36–75) | 61 (37–77) | 62 (38–77) | 63 (39–78) | 63 (40–78) | 65 (42–79) | 66 (44–79) | <0.001 |
| Age groups, n (%) | | | | | | | | | | | |
| Patient age 0–4 | 109 (1) | 147 (1) | 152 (1) | 200 (1) | 245 (1) | 261 (1) | 249 (1) | 203 (1) | 221 (1) | 234 (1) | 0.812 |
| Patient age 5–14 | 402 (4) | 572 (4) | 646 (4) | 771 (4) | 989 (4) | 1005 (4) | 889 (3) | 768 (4) | 785 (3) | 692 (3) | <0.001 |
| Patient age 15–24 | 1328 (13) | 1677 (12) | 1872 (11) | 2251 (11) | 2645 (10) | 2602 (10) | 2620 (10) | 1946 (9) | 2014 (9) | 1954 (8) | <0.001 |
| Patient age 25–34 | 1078 (10) | 1379 (10) | 1452 (9) | 1829 (9) | 2025 (8) | 1951 (7) | 1942 (7) | 1431 (7) | 1505 (6) | 1425 (6) | 0.027 |
| Patient age 35–44 | 1136 (11) | 1531 (11) | 1828 (11) | 2150 (10) | 2482 (10) | 2506 (9) | 2480 (9) | 1779 (8) | 1931 (8) | 1752 (8) | <0.001 |
| Patient age 45–54 | 1082 (10) | 1417 (10) | 1713 (10) | 2164 (10) | 2578 (10) | 2706 (10) | 2830 (11) | 2243 (11) | 2383 (10) | 2558 (11) | 0.004 |
| Patient age 55–64 | 1612 (15) | 1964 (14) | 2447 (15) | 2994 (14) | 3361 (13) | 3372 (13) | 3245 (12) | 2489 (12) | 2698 (12) | 2684 (12) | <0.001 |
| Patient age 65–74 | 1479 (14) | 1986 (14) | 2536 (15) | 3347 (16) | 3931 (15) | 4346 (16) | 4536 (17) | 3639 (17) | 3987 (17) | 4111(18) | 0.221 |
| Patient age ≥75 | 2370 (22) | 3184 (23) | 4187 (25) | 5614 (26) | 7300 (29) | 7878 (30) | 8061 (30) | 6569 (31) | 7723 (33) | 7925 (34) | <0.001 |
| Blunt injury, n (%) | 10,110 (95) | 13,246 (96) | 16,141 (96) | 20,438 (96) | 24,606 (96) | 25,687 (96) | 25,940 (97) | 20,386 (97) | 22,515 (97) | 22,654 (97) | <0.001 |
| Injury region, n (%) | | | | | | | | | | | |
| Polytrauma | 2303 (22) | 3071 (22) | 3584 (21) | 4143 (19) | 4613 (18) | 4733 (18) | 4859 (18) | 3998 (19) | 4268 (18) | 4150 (18) | <0.001 |
| Head injury with AIS ≥3 | 2307 (22) | 2859 (21) | 3546 (21) | 4412 (21) | 5067 (20) | 5400 (20) | 5443 (20) | 4334 (21) | 4747 (20) | 4974 (21) | 0.573 |
| Facial injury with AIS ≥3 | 18 (0.2) | 27 (0.2) | 39 (0.2) | 41 (0.2) | 55 (0.2) | 56 (0.2) | 56 (0.2) | 42 (0.2) | 50 (0.2) | 41 (0.2) | 0.881 |
| Neck injury with AIS ≥3 | 29 (0.3) | 44 (0.3) | 51 (0.3) | 60 (0.3) | 59 (0.2) | 61 (0.2) | 64 (0.2) | 64 (0.3) | 53 (0.2) | 52 (0.2) | 0.079 |
| Chest injury with AIS ≥3 | 1186 (11) | 1452 (10) | 1688 (10) | 2088 (10) | 2583 (10) | 2631 (10) | 2736 (10) | 2299 (11) | 2328 (10) | 2554 (11) | 0.206 |
| Abdominal and pelvic injury with AIS ≥3 | 274 (3) | 335 (2) | 367 (2) | 446 (2) | 513 (2) | 484 (2) | 433 (2) | 347 (2) | 350 (2) | 413 (2) | <0.001 |
| Spinal injury with AIS ≥3 | 600 (6) | 851 (6) | 1137 (7) | 1342 (6) | 1719 (7) | 1851 (7) | 1899 (7) | 1492 (7) | 1752 (8) | 1688 (7) | <0.001 |
| Upper extremity injury with AIS ≥3 | 277 (3) | 407 (3) | 520 (3) | 627 (3) | 806 (3) | 880 (3) | 872 (3) | 759 (4) | 700 (3) | 814 (3) | <0.001 |
| Lower extremity injury with AIS ≥3 | 1825 (17) | 2402 (17) | 3062 (18) | 3963 (19) | 4967 (19) | 5406 (20) | 5173 (19) | 3739 (18) | 4670 (20) | 4810 (21) | <0.001 |
| ISS, (median, IQR) | 14 (9–24) | 13 (9–24) | 13 (9–24) | 12 (9–21) | 10 (9–20) | 10 (9–20) | 11 (9–20) | 13 (9–21) | 11 (9–20) | 11 (9–20) | <0.001 |
| Injury severity groups, n (%) | | | | | | | | | | | |
| ISS 0–15 | 5595 (53) | 7374 (53) | 9006 (54) | 12,071 (57) | 14,974 (59) | 15,544 (58) | 15,539 (58) | 11,729 (56) | 13,303 (57) | 13,412 (57) | <0.001 |
| ISS 16–25 | 3044 (29) | 3923 (28) | 4761 (28) | 5788 (27) | 6785 (27) | 7223 (27) | 7230 (27) | 6004 (29) | 5403 (28) | 6484 (28) | 0.651 |
| ISS ≥26 | 1957 (18) | 2560 (18) | 3066 (18) | 3461 (16) | 3797 (15) | 3860 (15) | 4083 (15) | 3334 (16) | 3441 (15) | 3439 (15) | <0.001 |
| In-hospital mortality, n (%) | 1456 (13.7) | 1919 (13.9) | 2152 (12.8) | 2459 (11.5) | 2591 (10.1) | 2470 (9.3) | 2597 (9.7) | 2138 (10.2) | 2156 (9.3) | 2115 (9.1) | <0.001 |

IQR, Interquartile range; AIS, Abbreviated Injury Scale; ISS, Injury Severity Score

injured patients with ISS 0–15 and 16–25 was significantly lower than that of patients with ISS ≥16 (0.05 and 0.22 vs 1.00, p <0.001). The ISS cutoff value of ISS ≥16 has been commonly used as the definition of severe injury in Japanese trauma research [10–12, 14]. However, it may be necessary to change the definition of severe injury to include those having ISS cutoff values of ≥26 instead of ≥16, since trauma mortality trend has decreased and OR for mortality of injured patients with ISS ≥26 was relatively higher than those of patients with ISS 0–15 and 16–25 in the past decades.

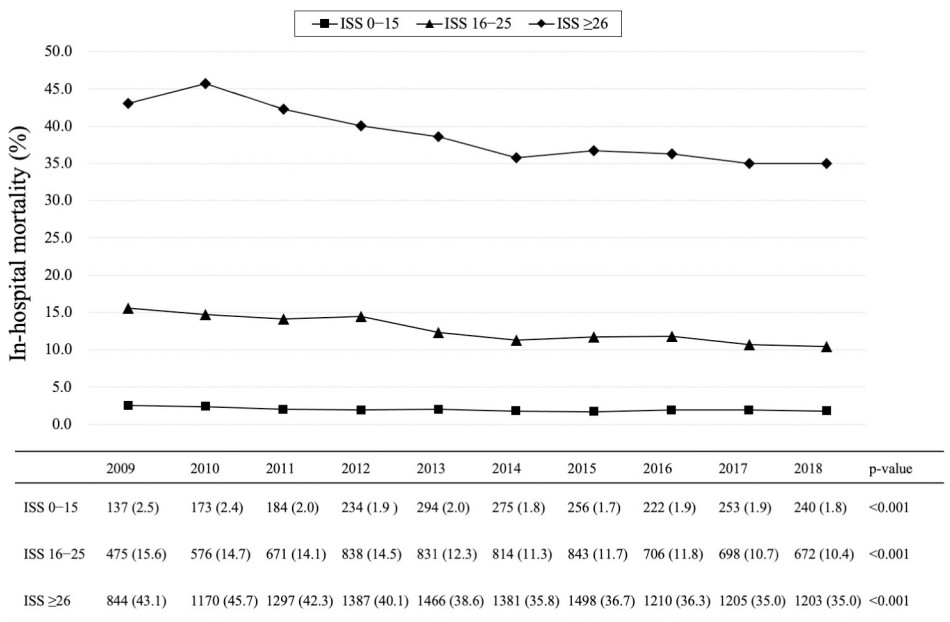

| | 2009 | 2010 | 2011 | 2012 | 2013 | 2014 | 2015 | 2016 | 2017 | 2018 | p-value |
|---|---|---|---|---|---|---|---|---|---|---|---|
| ISS 0−15 | 137 (2.5) | 173 (2.4) | 184 (2.0) | 234 (1.9 ) | 294 (2.0) | 275 (1.8) | 256 (1.7) | 222 (1.9) | 253 (1.9) | 240 (1.8) | <0.001 |
| ISS 16−25 | 475 (15.6) | 576 (14.7) | 671 (14.1) | 838 (14.5) | 831 (12.3) | 814 (11.3) | 843 (11.7) | 706 (11.8) | 698 (10.7) | 672 (10.4) | <0.001 |
| ISS ≥26 | 844 (43.1) | 1170 (45.7) | 1297 (42.3) | 1387 (40.1) | 1466 (38.6) | 1381 (35.8) | 1498 (36.7) | 1210 (36.3) | 1205 (35.0) | 1203 (35.0) | <0.001 |

**Fig 2. In-hospital mortality trends among injured patients according to ISS groups.**

In this study, there were specific injured patients with a high mortality risk in whom the mortality trend also remained unimproved from 2009 to 2018. First, the mortality risk and trend of injured patients varied significantly according to age (S2 Table and Table 2). In patients aged ≥15 years, the mortality risk increased steadily with increasing age, and the mortality rate of elderly patients with ISS ≥26 remained high despite the improvement in mortality trends of elderly patients with ISS ≥16. Japan has the most rapidly increasing number of aging citizens in the world, and the proportion of elderly patients aged > 75 years with the highest mortality risk accounted for 34% of this study cohort in 2018. As Japan is expecting a rapid change in its population structure with a growing elderly population, it is important to establish a trauma care system suitable for such patients to improve mortality by considering their age-related characteristics and various mortality risks. In patients aged younger than 15 years, the mortality trend in injured patients with ISS ≥26 did not significantly improve, and the in-hospital mortality risk of patient aged 0−4 years with ISS ≥26 remained high during the 10 years of study (26.1%−58.3%). Previous studies suggested that injured patients younger than 5 years had a higher mortality risk, because they included a high proportion of severe head injuries [11, 15]. Our results also showed that the OR for mortality in patients with head injury and AIS ≥3 was 3.48. Therefore, it may be important to establish the therapeutic strategies suitable for patients with severe head injuries to decrease the mortality risk of younger injured patients in Japan. Second, our results showed that the mortality risk of patients with penetrating injury was 1.7 times higher than that of patients with blunt injury (Table 2), as shown in a previous study that used JTDB data from 2004 to 2011. The mortality trend in patients with penetrating injuries with ISS 0−15 and ≥26 did not decrease significantly, and the mortality rate of patients with penetrating injuries with ISS ≥26 remained extremely high from 40.0% to 51.3%, despite the improvements seen in blunt injury patients regardless of the anatomical severity. Furthermore, in patients with a significantly high mortality risk due to polytrauma or injury regions with AIS ≥3, such as head, neck, chest, abdominal and pelvic, or lower extremity injury, the mortality trends in those with neck injury and abdominal/pelvic injury and ISS

**Table 2. Multivariable logistic regression analysis of in-hospital mortality among injured patients.**

| | All patients (n = 209,290) | | |
| --- | --- | --- | --- |
| | OR | 95% CI | p value |
| Admission year groups | | | |
| 2009 | 1.63 | (1.502–1.763) | <0.001 |
| 2010 | 1.66 | (1.544–1.791) | <0.001 |
| 2011 | 1.48 | (1.377–1.587) | <0.001 |
| 2012 | 1.39 | (1.301–1.492) | <0.001 |
| 2013 | 1.26 | (1.174–1.343) | <0.001 |
| 2014 | 1.11 | (1.035–1.185) | 0.003 |
| 2015 | 1.12 | (1.052–1.202) | 0.001 |
| 2016 | 1.12 | (1.042–1.199) | 0.002 |
| 2017 | 1.05 | (0.976–1.122) | 0.201 |
| 2018 | 1.00 | | – |
| Age groups | | | |
| Patient age 0–4 | 0.43 | (0.351–0.518) | <0.001 |
| Patient age 5–14 | 0.20 | (0.175–0.230) | <0.001 |
| Patient age 15–24 | 0.39 | (0.369–0.418) | <0.001 |
| Patient age 25–34 | 0.49 | (0.462–0.526) | <0.001 |
| Patient age 35–44 | 0.51 | (0.483–0.544) | <0.001 |
| Patient age 45–54 | 0.53 | (0.497–0.557) | <0.001 |
| Patient age 55–64 | 0.56 | (0.530–0.588) | <0.001 |
| Patient age 65–74 | 0.66 | (0.634–0.696) | <0.001 |
| Patient age ≥75 | 1.00 | | – |
| Injury mechanism | | | |
| Blunt injury | 0.59 | (0.533–0.652) | <0.001 |
| Penetrating injury | 1.00 | | – |
| Injury region | | | |
| Polytrauma | 2.95 | (2.618–3.336) | <0.001 |
| Head injury with AIS ≥3 | 3.48 | (3.094–3.913) | <0.001 |
| Facial injury with AIS ≥3 | 1.11 | (0.492–2.524) | 0.794 |
| Neck injury with AIS ≥3 | 8.73 | (6.708–11.350) | <0.001 |
| Chest injury with AIS ≥3 | 3.13 | (2.775–3.536) | <0.001 |
| Abdominal and pelvic injury with AIS ≥3 | 3.90 | (3.326–4.563) | <0.001 |
| Spinal injury with AIS ≥3 | 0.82 | (0.704–0.944) | 0.006 |
| Upper extremity injury with AIS ≥3 | 0.76 | (0.572–1.001) | 0.051 |
| Lower extremity injury with AIS ≥3 | 1.48 | (1.313–1.674) | <0.001 |
| Injury severity groups, n (%) | | | |
| ISS 0–15 | 0.05 | (0.045–0.051) | <0.001 |
| ISS 16–25 | 0.22 | (0.206–0.224) | <0.001 |
| ISS ≥26 | 1.00 | | – |

OR, odds ratio; CI, confidence interval; AIS, Abbreviated Injury Scale; ISS, Injury Severity Score.

≥26 did not improve. A previous study suggested that the survival benefit in patients with severe head injury, penetrating injury, pelvic injury, and solid organ injury may be improved by centralization to trauma centers with a higher quality of trauma care [16]. Several studies have also suggested that centralizing patients with penetrating injury or pediatric injured patients to higher-volume hospitals may contribute to the survival benefit [17, 18]. Therefore,

it is necessary to achieve a trauma care system that centralizes specific severely injured patients to hospitals with a high volume and quality to improve their outcomes, based on the results of a nationwide clinical research based on the injury mechanism and region.

Our study had some limitations. First, there was a selection bias, as not all Japanese hospitals participated in the JTDB registry, and the number of participating hospitals varied across the 10-year study period. Moreover, there were missing data in the JTDB dataset. Second, the results of the trend test analyzing the small study cohort resulted in biased conclusions. Third, with regards to the trauma score for predicting injury severity and mortality, several studies have reported that are based on not only anatomical severity, but also on physiological factors and results of blood test, which may be more useful and accurate to define severe injury in patients with high mortality risk [6, 13, 18]. In the future, it is necessary to evaluate not only anatomical severity but physiological variables and examination results as the predictor of trauma mortality.

## Conclusions

From 2009 to 2018, the in-hospital mortality trend among all injured patient groups with ISS 0–15, 16–25, and ≥26 showed significant decreases in Japan. Differences were noted in mortality trends and risks according to age, injury mechanism, injury region, and anatomical injury severity. For specific patients with older age, those with penetrating injury, and/or those with specific injury regions with AIS ≥3, it may be necessary to change the definition of these patients according to trauma mortality.

## Supporting information

**S1 File.**
(XLSX)

**S1 Table. Number of patients with missing data by study year.** Some fitted into more than one variable.
(DOCX)

**S2 Table. In-hospital mortality trends among injured patients by age groups and ISS groups.** ISS, Injury Severity Score.
(DOCX)

**S3 Table. In-hospital mortality trends by injury mechanism and ISS groups.** ISS, Injury Severity Score.
(DOCX)

**S4 Table. In-hospital mortality trends by injury region and ISS groups.** AIS, Abbreviated Injury Scale; ISS, Injury Severity Score.
(DOCX)

## Author Contributions

**Conceptualization:** Chiaki Toida, Takashi Muguruma.

**Data curation:** Chiaki Toida, Masayasu Gakumazawa, Mafumi Shinohara, Takeru Abe.

**Formal analysis:** Chiaki Toida, Takeru Abe.

**Funding acquisition:** Chiaki Toida.

**Investigation:** Chiaki Toida, Takashi Muguruma, Mafumi Shinohara.

**Methodology:** Chiaki Toida, Takashi Muguruma.

**Resources:** Chiaki Toida.

**Software:** Takeru Abe.

**Supervision:** Ichiro Takeuchi.

**Validation:** Chiaki Toida, Takashi Muguruma, Masayasu Gakumazawa, Takeru Abe.

**Visualization:** Chiaki Toida, Mafumi Shinohara, Takeru Abe.

**Writing – original draft:** Chiaki Toida.

**Writing – review & editing:** Takashi Muguruma, Masayasu Gakumazawa, Mafumi Shinohara, Takeru Abe, Ichiro Takeuchi.

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
