## [Decision Letter · Decision Letter 0]

22 Apr 2022

PONE-D-22-05214Ten-year in-hospital mortality trends among Japanese injured patients by age, injury severity, injury mechanism, and injury region: A nationwide observational studyPLOS ONE

Dear Dr. Toida,

Thank you for submitting your manuscript to PLOS ONE. After careful consideration, we feel that it has merit but does not fully meet PLOS ONE’s publication criteria as it currently stands. Therefore, we invite you to submit a revised version of the manuscript that addresses the points raised during the review process.

ACADEMIC EDITOR: 

Please addressed questions and comments raised by the reviewer 1, especially related to abstract and methods. 

Data should be made available as per PLOS's data policy.

We look forward to receiving your revised manuscript.

Kind regards,

Tze-Woei Tan, M.D.

Academic Editor

PLOS ONE

Journal Requirements:

“C.T. received a grant from the General Insurance Association of Japan [Grant No.21-08].”

Reviewers' comments:

Reviewer's Responses to Questions

**Comments to the Author**

1. Is the manuscript technically sound, and do the data support the conclusions?

Reviewer #1: Partly

Reviewer #2: Yes

2. Has the statistical analysis been performed appropriately and rigorously? 

Reviewer #1: Yes

Reviewer #2: Yes

3. Have the authors made all data underlying the findings in their manuscript fully available?

Reviewer #1: No

Reviewer #2: No

4. Is the manuscript presented in an intelligible fashion and written in standard English?

Reviewer #1: Yes

Reviewer #2: Yes

5. Review Comments to the Author

Reviewer #1: The Injury Severity Score (ISS) is an established scoring system for accessing the severity of injury across the world. In Japan, ISS score > 16 is used to define severe trauma. Authors performed a 10 year analysis retrospective analysis of the Japanese Trauma National Bank (JTNB) stratifying patient based on ISS score. They determined the mortality over the years stratifying by ISS score and found a decreasing trend in the mortality and higher mortality with higher ISS score. Based on the mortality rate, they recommended using a higher ISS score cut off to define severely injured trauma patients. The authors do good job in the study to help answer the question they sought out to answer however, I have a few questions and revision regarding the study

P2 L24: The authors mention ISS Score > 16 is used to define severe trauma in Japan. In the United States and across the world, ISS Score of > 15 is used to define severe trauma. Can you clarify why this difference

P2 L28: The aim cannot be to validate. To validate, you need a different dataset.

P2 L29: Please write down the full form of JTNB

P2: The abstract is missing methods section and details of the study as the authors jump into results

P2: L32: Please add p value

P2: L33-37 can be deleted and kept for the manuscript. It confuses the reader and not relevant for the aim

P2: L40: The conclusion needs to be what they find and not another hypothesis. The found lower mortality in the ISS group 16-25 and has been decreasing. Based on this they suggest a change in cut off to ISS score of > 26

P3 L58-60: The authors mention ISS Score > 16 is used to define severe trauma in Japan. In the United States and across the world, ISS Score of > 15 is used to define severe trauma. Can you clarify why this difference

P3: L63-65: The mortality has decreased and other factors like mentioned by authors are contributing. Multiple scoring systems have been used like Revised Trauma Score (RTS), Trauma Injury Severity Score (TRISS) and New Injury Score. What do the authors think about comparing ISS with TRISS which has becoming the standard scoring system.

P4: L69-71: Please edit the aim as per the abstract

P4: L86-88: Can you please include the excluded patient each year. It would be good to know that so missing bias may be limited. Do not want a lot of patient missing in the earlier years compared to recent years. I would anticipate the proportion of missing patients be equal across the years.

P4: Do you have GSC score or patient admission BP and HR data? That may be useful to add

P4: L88: The figure 1 is excellent and adding excluded patient based on year will be helpful

P4L L92: Please change sex to gender

P5: L95: Why divide into 9 groups age groups? Is that the way data reported in the JTNB?

P5: A very good description of the methods

P6: Ethics statement can be moved after study settling and patient poplutation

P6: Result paragraph one, can some of the statement deleted. Most of the information is in the table

P7-14: Please edit result. Focus on the aim result decrease mortality with increasing ISS. The stratification by age, AIS, blunt etc

P7: May be a figure demonstrating Mortality rate and ISS score will help to make the point stronger. The table looks overwhelming

P9: Table 2 although a lot of work, does not add more much to the aim of the study

P12: Table 4 although a lot of work, does not add more much to the aim of the study

P11-12: May be change the table 3 and 4 into figures or have them as supplement

P15: L188-194: Please edit to focus on 1: decrease mortality with increasing ISS. The age, AIS etc are all subanalysis.

P15: L195-202: the paragraph does not add much context to the study. I would add a paragraph focusing on the main result regarding ISS and mortality

P16: L219: Please move this paragraph above the paragraph about the age and mortality as that was not the focus. Also please edit to clear to focus how it aligns with your study

P17: L238-: please move this paragraph as a second paragraph as this is the focus

P17: L250: There are more limitations to the study like GCS, vitals not recorded, missing data, not compared to other major scoring system like TRISS

P17: L261: I think defining ISS > 26 as the new cut off without validation is a strong statement. May be it can be a suggestion or more studies need to be performed.

6. PLOS authors have the option to publish the peer review history of their article (what does this mean?). If published, this will include your full peer review and any attached files.

Reviewer #1: No

Reviewer #2: No

---

## [Author Response · Author response to Decision Letter 0]

6 Jun 2022

June 6, 2022

Dr. Tze-Woei Tan, M.D.,

Academic Editor

PLOS ONE 

Dear Editors,

On behalf of all the authors, I would like to thank you for your response and we thank the reviewers for their comments regarding our manuscript (PONE-D-22-05214) titled “Ten-year in-hospital mortality trends among Japanese injured patients by age, injury severity, injury mechanism, and injury region: A nationwide observational study.” We hope that the revised version of the manuscript will be moved closer to publication in PLOS ONE.

The manuscript has been modified in accordance with the extensive and insightful comments from the reviewers. The point-by-point responses to all comments have been prepared and given below. The revised manuscript is attached herewith. Please find the modifications in the revised manuscript marked in red font. Moreover, please permit that the number of words of the abstract exceeded the recommended limit because of having revised it according to reviewers’ comment.

C.T. received a grant from the General Insurance Association of Japan [Grant No.21-08]. The funders had no role in this study design, data collection and analysis, decision to publish, or preparation of the manuscript.

Please do not hesitate to contact us for further clarifications regarding our manuscript. Thank you for your consideration. I look forward to hearing from you.

Sincerely,

Chiaki Toida

Department of Disaster Medical Management, University of Tokyo

7-3-1 Hongo, Bunkyo-ku, Tokyo 113-8655, Japan

Phone: +81-3-3815-5411

Fax: +81-3-3814-6446

E-mail: toida-ygc@umin.ac.jp

 

Reviewer’s comments (in blue) and Answers (in black) follow.

We wish to express our deep appreciation for the valuable comments from the Reviewer regarding our manuscript. We believe that our manuscript has benefited greatly from these comments.

Reviewer: 1

1. P2 L24: The authors mention ISS Score > 16 is used to define severe trauma in Japan. In the United States and across the world, ISS Score of > 15 is used to define severe trauma. Can you clarify why this difference 

Response: We thank the Reviewer for this comment. We used patients with ISS ≥16 and ISS >15 as the same meaning, because the groups were divided into three, based on injury severity (ISS 0−15, 16−25, and ≥26). We included the definition of severe trauma as ISS ≥ 16 not ISS > 15.

2. P2 L28: The aim cannot be to validate. To validate, you need a different dataset. 

Response: We thank the Reviewer for this suggestion. We agree with this suggestion. We revised the aim of this study in the Abstract as follows.

This study aimed to analyze the prevalence, in-hospital mortality, and odds ratio (OR) for mortality in Japanese injured patients by age, injury mechanism, injury region, and injury severity over 10 years.

3. P2 L29: Please write down the full form of JTNB

Response: Thank you for your comment. JTDB has been spelled out in the Abstract as follows.

the Japan Trauma Data Bank (JTDB)

4. P2: The abstract is missing methods section and details of the study as the authors jump into results 

Response: We thank the Reviewer this suggestion. We added the details of this study method in the Abstract as follows.

This study evaluated 10-year trends of the prevalence and in-hospital mortality and risk factors associated with in-hospital mortality.

5. P2: L32: Please add p value

Response: As you pointed out, we added p value as p < 0.001.

6. P2: L33-37 can be deleted and kept for the manuscript. It confuses the reader and not relevant for the aim 

Response: We thank the Reviewer for this valuable suggestion. We strongly agree with your comment. We deleted these sentences.

7. P2: L40: The conclusion needs to be what they find and not another hypothesis. The found lower mortality in the ISS group 16-25 and has been decreasing. Based on this they suggest a change in cut off to ISS score of > 26 

Response: We thank the Reviewer suggestion. We revised the Conclusion according to your comment 7 and 24 as follows.

From 2009 to 2018, the in-hospital mortality trend among all injured patient groups with ISS 0–15, 16–25, and ≥26 showed significant decreases in Japan. Differences were noted in mortality trends and risks according to anatomical injury severity.

8. P3 L58-60: The authors mention ISS Score > 16 is used to define severe trauma in Japan. In the United States and across the world, ISS Score of > 15 is used to define severe trauma. Can you clarify why this difference

Response: As mentioned above, we used patients with ISS ≥ 16 and ISS > 15 as the same meaning. Because we split into three groups based on injury severity (ISS 0−15, 16−25, and ≥ 26), we mentioned the definition of severe trauma with ISS ≥16 not ISS >15.

9. P3: L63-65: The mortality has decreased and other factors like mentioned by authors are contributing. Multiple scoring systems have been used like Revised Trauma Score (RTS), Trauma Injury Severity Score (TRISS) and New Injury Score. What do the authors think about comparing ISS with TRISS which has becoming the standard scoring system. 

Response: We thank the Reviewer for this valuable comment. In Japan, Epidemiological trauma outcome research based on the JTDB data also frequently used an ISS cutoff value of ≥16 as the definition of severe injury [10–12]. However, there is no long-term study evaluating the mortality trends and risk of injured patients in Japanese cohort by detailed classification of injury severity. Therefore, we aimed to analyze the prevalence, in-hospital mortality, and odds ratio (OR) for mortality in Japanese injured patients by anatomical injury severity, which is one of the predictors used to calculate the probability of survival by TRISS method. We revised the Introduction section as follows.

The mortality of patients with trauma has decreased in recent years due to the development of the trauma care system [9–11]. Moreover, the mortality trend and risk of severely injured patients vary widely according to age, injury mechanism, injury region, and/or injury severity [3, 10–13]. However, to the best of our knowledge, no long-term study has evaluated the mortality trends and risks of injured patients in a Japanese cohort using detailed classification of age and/or injury severity. Therefore, this 10-year nationwide study aimed to analyze the prevalence, in-hospital mortality, and odds ratio (OR) for mortality in Japanese injured patients by age and injury severity including injury mechanism and injury region.

10. P4: L69-71: Please edit the aim as per the abstract 

Response: We thank the Reviewer for this suggestion. We agree to this suggestion. We revised the aim of this study in the Text as follows.

Therefore, this 10-year nationwide study aimed to analyze the prevalence, in-hospital mortality, and odds ratio (OR) for mortality in Japanese injured patients by age and injury severity including injury mechanism and injury region.

11. P4: L86-88: Can you please include the excluded patient each year. It would be good to know that so missing bias may be limited. Do not want a lot of patient missing in the earlier years compared to recent years. I would anticipate the proportion of missing patients be equal across the years.

Response: We thank the Reviewer for this comment. We showed the proportion of missing data across the study year in S1 Table, because it is difficult to show these data in Figure 1. Regrettably, there are not differences in the proportion of missing data by study years.

12. P4: Do you have GSC score or patient admission BP and HR data? That may be useful to add 

Response: As you pointed out, we analyzed the data of GCS score, systolic BP, and HR at hospital admission. We revised the Method and Result section as follows.

Method

We collected information on the following variables from the JTDB: demographic characteristics (age in years, gender, year of hospital admission); clinical parameters (injury mechanism, AIS of the injured region, ISS, Glasgow Coma Scale (GCS) score, systolic blood pressure (sBP), and heart rate (HR) at hospital admission);…

Result

The median GCS score, sBP, and HR at hospital admission was 15 (IQR, 13–15), 135 (IQR, 114–157), and 20 (IQR, 16–24), respectively.

13. P4: L88: The figure 1 is excellent and adding excluded patient based on year will be helpful 

Response: As mentioned above, we showed the proportion of missing data across the study year in S1 Table.

14. P4L L92: Please change sex to gender

Response: As you pointed out, we revised sex in gender.

15. P5: L95: Why divide into 9 groups age groups? Is that the way data reported in the JTNB? 

Response: Owing to aging issues in Japan, the definition of the elderly was classified into two groups; early-stage elderly (person between 65-74 years old) and later-stage elderly (person over 75 years old). The annual report showed that these two groups have differences in the mortality and morbidity.

https://www8.cao.go.jp/kourei/english/annualreport/2021/pdf/2021.pdf

Moreover, in children, previous study showed that preschool children had higher mortality than the other age groups.

J Clin. Med. 2021, Clin. Med. 2021, 10, 1072. https://doi.org/ 10.3390/jcm10051072

Therefore, we divided in 9 age groups (0−4, 5−14, 15−24, 25−34, 35−44, 45−54, 55−64, 65−74, and ≥75 years old) in this study, as it is predicted that the mortality rates are different by age group. 

16. P5: A very good description of the methods 

Response: Thank you for your kindly comment. 

17. P6: Ethics statement can be moved after study settling and patient poplutation

Response: As you pointed out, we moved the ethics statement to after the study setting and patients population.

18. P6: Result paragraph one, can some of the statement deleted. Most of the information is in the table

Response: We thank the Reviewer comment. We delated the most of information including the table 1 as follows.

A total of 201,723 patients (96%) had a blunt injury. The number of patients with two or more injury regions with AIS ≥ 3 was as follows: polytrauma (n = 39,722, 19%); head injury (n = 43,089, 21%); facial injury (n = 425, 0.2%); neck injury (n = 537, 0.3%); chest injury (n = 21,545, 10%); abdominal and pelvic injury (n = 3962, 2%); spinal injury (n = 14,331, 7%); upper extremity injury (n = 6662, 3%); lower extremity injury (n = 40,017, 19%).

19. P7: May be a figure demonstrating Mortality rate and ISS score will help to make the point stronger. The table looks overwhelming

Response: As you pointed out, we demonstrated 10-years mortality trends by ISS groups in Figure 2. 

20. P7-14: Please edit result. Focus on the aim result decrease mortality with increasing ISS. The stratification by age, AIS, blunt etc

P9: Table 2 although a lot of work, does not add more much to the aim of the study

P12: Table 4 although a lot of work, does not add more much to the aim of the study

P11-12: May be change the table 3 and 4 into figures or have them as supplement

Response: We thank the Reviewer valuable comments. We agree that we focus on the mortality trend by ISS groups, decreasing the quantity of data presentation. Therefore, we added the Figure 2 which showed the mortality trends by ISS groups, and changed Table 2, 3 ,4 to Supplement Table 2, 3, 4. We revised the results as follows.

The in-hospital mortality trends among injured patients by age group, injury mechanism, and injury region according to three groups based on injury severity (ISS 0−15, 16−25, and ≥26) are shown in Table S2, S3, and S4. Among injured patients with age >15 years, in-hospital mortality of patients with ISS 16–25 and ≥26 significantly decreased over the 10-year study period (Table S2). The in-hospital mortality among patients with blunt injury showed a significant decrease in all ISS groups (Table S3). Moreover, in patients with ISS from 16–25, the in-hospital mortality of patients with polytrauma, head, chest, abdominal and pelvic, spinal, upper extremity, and lower extremity injuries with AIS ≥3 showed a significant decrease (Table S4).

22. P15: L188-194: Please edit to focus on 1: decrease mortality with increasing ISS. The age, AIS etc are all subanalysis. 

Response: As you pointed out, we revised the Discussion as follows.

This 10-year nationwide study in Japan showed that the in-hospital mortality trend significantly decreased in all injured patient groups with ISS 0–15, 16–25, and ≥26. However, there were differences in the mortality trends and risk according to the age, injury mechanism, injury region, and anatomical severity. Moreover, the in-hospital mortality and OR for mortality in the patient group with ISS ≥26 were higher than those in two patient groups with ISS 0−15 and 16−25.

23. P15: L195-202: the paragraph does not add much context to the study. I would add a paragraph focusing on the main result regarding ISS and mortality

P16: L219: Please move this paragraph above the paragraph about the age and mortality as that was not the focus. Also please edit to clear to focus how it aligns with your study

P17: L238-: please move this paragraph as a second paragraph as this is the focus

Response: We thank the Reviewer for valuable suggestion. We agree with your suggestion. We deleted the second paragraph and revised the Discussion to focus on the main results regarding ISS and mortality by changing the order of paragraphs.

24. P17: L250: There are more limitations to the study like GCS, vitals not recorded, missing data, not compared to other major scoring system like TRISS 

Response: As you pointed out, we added the limitation about other major scoring systems, including physiological factors and/or results of blood test as follows.

Third, with regards to the trauma score for predicting injury severity and mortality, several studies have reported that are based on not only anatomical severity, but also on physiological factors and results of blood test, which may be more useful and accurate to define severe injury in patients with high mortality risk [6,13,18]. In the future, it is necessary to evaluate not only anatomical severity but physiological variables and examination results as the predictor of trauma mortality.

25. P17: L261: I think defining ISS > 26 as the new cut off without validation is a strong statement. May be it can be a suggestion or more studies need to be performed. 

Response: We thank the Reviewer for this valuable suggestion. We revised the Conclusion as follows.

From 2009 to 2018, the in-hospital mortality trend among all injured patient groups with ISS 0–15, 16–25, and ≥26 showed significant decreases in Japan. Differences were noted in mortality trends and risks according to age, injury mechanism, injury region, and anatomical injury severity. For specific patients with older age, those with penetrating injury, and/or those with specific injury regions with AIS ≥3, it may be necessary to change the definition of these patients according to trauma mortality.

---

## [Decision Letter · Decision Letter 1]

22 Jul 2022

Ten-year in-hospital mortality trends among Japanese injured patients by age, injury severity, injury mechanism, and injury region: A nationwide observational study

PONE-D-22-05214R1

Dear Dr. Toida,

We’re pleased to inform you that your manuscript has been judged scientifically suitable for publication and will be formally accepted for publication once it meets all outstanding technical requirements.

Kind regards,

Tze-Woei Tan, M.D.

Academic Editor

PLOS ONE

Reviewers' comments:

Reviewer's Responses to Questions

**Comments to the Author**

1. If the authors have adequately addressed your comments raised in a previous round of review and you feel that this manuscript is now acceptable for publication, you may indicate that here to bypass the “Comments to the Author” section, enter your conflict of interest statement in the “Confidential to Editor” section, and submit your "Accept" recommendation.

Reviewer #1: All comments have been addressed

Reviewer #2: All comments have been addressed

2. Is the manuscript technically sound, and do the data support the conclusions?

Reviewer #1: Yes

Reviewer #2: Yes

3. Has the statistical analysis been performed appropriately and rigorously? 

Reviewer #1: Yes

Reviewer #2: Yes

4. Have the authors made all data underlying the findings in their manuscript fully available?

Reviewer #1: No

Reviewer #2: Yes

5. Is the manuscript presented in an intelligible fashion and written in standard English?

Reviewer #1: Yes

Reviewer #2: Yes

6. Review Comments to the Author

Reviewer #1: The authors have addressed all the concerns raised and provided satisfactory explanation for the questions. They have revised the manuscript based on the concerns. The manuscript reads better. I have no further questions.

Reviewer #2: (No Response)

7. PLOS authors have the option to publish the peer review history of their article (what does this mean?). If published, this will include your full peer review and any attached files.

Reviewer #1: No

Reviewer #2: **Yes: **Navdeep Samra, MD, FACS

---

## [Editor Report · Acceptance letter]

12 Aug 2022

PONE-D-22-05214R1 

Ten-year in-hospital mortality trends among Japanese injured patients by age, injury severity, injury mechanism, and injury region: A nationwide observational study 

Dear Dr. Toida:

I'm pleased to inform you that your manuscript has been deemed suitable for publication in PLOS ONE. Congratulations! Your manuscript is now with our production department. 

Kind regards, 

on behalf of

Dr. Tze-Woei Tan 

Academic Editor

PLOS ONE